# COVID-19 and Its Impact on Upper Gastrointestinal (GI) Cancer Management

**DOI:** 10.3390/cancers13030397

**Published:** 2021-01-21

**Authors:** Shalini Fernando, Mesel Veli, Borzoueh Mohammadi, Andrew Millar, Khurum Khan

**Affiliations:** 1North Middlesex University Hospital, Sterling Way, London N181QX, UK; shalini.fernando2@nhs.net (S.F.); mesel.veli@nhs.net (M.V.); andrewmillar@nhs.net (A.M.); 2Department of GI Oncology, University College Hospital, 235 Euston Road, Bloomsbury, London NW1 2BU, UK; borzoueh.mohammadi@nhs.net; 3UCL Cancer Trials Centre 72 Huntley Street, Bloomsbury, London WC1E 6DD, UK

**Keywords:** gastrointestinal, cancer, coronavirus, endosocopy, COVID-19, SARS-Cov-2

## Abstract

**Simple Summary:**

In the midst of a pandemic resulting from the new virus, severe acute respiratory syndrome coronavirus 2, we carried out a review of the impact of the virus on patients suffering with upper gastrointestinal cancers. This previously unknown, but highly infectious virus, has rapidly spread throughout the world, causing devastation to people’s health on a global scale. The scientific and medical community have had to adapt and learn how to manage the virus, which has had knock on effects to patients suffering other diseases. Health services have been severely disrupted, so we reviewed the impact of this, specifically relating to the diagnosis and treatment of patients with upper gastrointestinal cancers. The situation is rapidly changing; therefore, we share the findings of our critical analysis of the available literature.

**Abstract:**

Coronavirus disease 2019 (COVID-19), caused by the novel, severe acute respiratory syndrome coronavirus 2 (SARS-CoV-2) virus, has left dramatic footprints on human health and economy. Cancer, whilst not an infective disease, is prevalent in epidemic proportions and cannot be pretermitted due to the impact of COVID-19. As we emanate from the second national lockdown in the UK with mixed feelings of hope and despair—due to vaccination and new COVID-19 variant, respectively—we reflect on the impact of the first wave on the provision on diagnosis and management of with upper gastrointestinal (UGI) cancers. This review provides a critical analysis of available literature on COVID-19 and its impact on cancer management in general and that of UGI cancers in particular.

## 1. Background

Coronavirus disease 2019 (COVID-19), caused by the novel, severe acute respiratory syndrome coronavirus 2 (SARS-CoV-2) virus, was declared a pandemic by the World Health Organisation in March 2020. Cancer, whilst not an infective disease, is prevalent in epidemic proportions and should not feasibly be side-lined by the management of the public health emergency, caused by COVID-19. As we emerge from the second National lockdown in the UK, we reflect on the impact of the first wave on the provision of treatment for patients diagnosed with upper gastrointestinal (UGI) cancers; hopeful that lessons learned will help us navigate through any further surges in virus cases, protecting the delivery of cancer care.

From the outset, cancer patients were identified as being more vulnerable to the virus so were placed on the list of people asked to shield at home. This was entirely reasonable, given that cancer patients are often treated with chemotherapy with recognised myelosuppressive effects [1]. Early data from China supported this notion [1,2]. The National Clinical Research Center for Respiratory Disease in conjunction with the National Health Commission of the People’s Republic of China published a nationwide review of 1590 people infected with COVID-19, which revealed that cancer patients, who made up (1%) of the prospectively analysed cohort, were more likely to have a severe event, HR 5.34, 95% CI 1.8–16.18. Cancer patients (*n* = 28), who had received their last systemic anticancer treatment (SACT) within 14 days had increased risk of developing severe events (HR = 4.08, 95% CI 1.09–15.32) [3]. A severe event was defined as admission to an Intensive Care Unit (ICU), requiring ventilation and death. Time to deterioration was more rapid in cancer patients compared to non-cancer patients, median 13 days; however, this was a non-specific and underpowered study. Numerous other descriptive or retrospective clinical studies examining patient outcomes in China, from the onset of the pandemic, suggested that age and medical co-morbidities, including cancer, increased the risk of Sars-Cov-2 virus infection [1,2,4] and, in severe cases, death resulted from multiorgan failure and acute respiratory distress syndrome [4,5,6]. The first national lockdown in the UK commenced on 23rd March 2020, with the aim of saving lives by ensuring the National Health Service (NHS) was not overwhelmed by the burden of treating uncontrolled numbers of patients infected with COVID-19. The immediate focus was to treat infected patients; clinical presentation varying from those suffering cough and fever, to patients needing hospital admission for oxygen therapy, up to the sickest patients who required admission to ICU for ventilation. As a result, hospital services and staff were redeployed with the principal aim of improving ICU capacity in order to meet the unprecedented demands. Non-emergency clinical services, including diagnostics and elective surgery, were de-prioritised, an inevitable consequence of which was compromised cancer care [7,8]. Simultaneously, population isolation, shielding and lockdowns have had a considerable impact on the numbers of patients presenting with symptoms. Subsequently, there has been a massive fall in referrals of symptomatic patients from primary into secondary care [7]. Figure 1 demonstrates the descent in referrals from primary to secondary care for all cancers at the height of the pandemic.

## 2. Upper GI Endoscopies and Diagnostic Challenges

An average of 35,478 endoscopy procedures were performed every week in the pre-COVID era in the UK [9]. Following the UK government’s guidance to mitigate the risk of further spreading SARS-CoV-2 via aerosol droplets, all non-emergency endoscopic procedures in the UK were stopped for six weeks from the 23rd March 2020, on the advice of British Society and Gastroenterology (BSG) and the Joint Advisory group (JAG). Patients were consequently divided into three groups:Emergency/essential, so should be done;Defer until further notice;Needs discussion (at Consultant level).

Urgent suspected cancer (USC) or 2-week wait (2WW) referrals and endoscopic ultrasound (EUS) for cancer staging both fell into category 3. Therefore, all suspected cancer referrals were scrutinised to ensure endoscopy was reserved for those truly in need, taking into account the availability of alternative staging procedures, availability of surgical services and whether endoscopy would change management [10].

During the pandemic, only 12% of the pre-COVID volume of procedures was performed, with a mere 5% of pre-COVID volume occurring at the height of the pandemic (end of March). By the end of May 2020, the number of procedures had risen; however, it was still only 20% of pre-COVID activity. Subsequently, the number of cancer cases detected plummeted as well; it decreased from an average of 677 per week in the pre-COVID era to 283 per week in the COVID era [9]. Moreover, oesophageal and gastric cancers presented late and were more likely to be of advanced stage at diagnosis, potentially leading to adverse outcomes [11]. Furthermore, due to the reductions in diagnostic endoscopies, a large proportion of patients will remain undiagnosed [11]. 

## 3. Changes to Surgical Practices—Operating Incognito 

New patient pathways were mandated in order to be able to deliver cancer care safely during the ongoing pandemic. At the start of the pandemic, the risks of operating were not known [12]. The Royal College of Radiologists published a document on 25 March 2020 entitled “Considerations for treatment of oesophagogastric cancers within the United Kingdom during the Covid-19 pandemic”. The treatment advice document was devised after seeking the expert opinion of cancer specialists. Prioritisation protocols were developed and published to guide the prioritisation of elective surgery. They were divided into four categories (Figure 2):

Priority 1—needing urgent/emergency surgery.Priority 2—cases able to be deferred within 4 weeks.

Additionally, they decreed that all patients undergoing elective surgery would have to self-isolate for 14 days, complete a COVID-19 risk assessment and have a negative Sars-CoV-2 swab prior to admission. CT chest was required immediately pre-operatively. Other measures undertaken included deferring laparoscopic staging until the completion of neoadjuvant chemotherapy, assessing those already on curative pathways on the completion of neoadjuvant chemotherapy and identifying patients who could be managed endoscopically, e.g., T1a and T1b oesophageal and gastric tumours. Furthermore, to reduce endoscopy demands, patients presenting with painless jaundice, obstructed ducts and pancreatic masses were triaged straight to surgery. Liver resections for the management of hepatocellular carcinoma (HCC) went ahead, given there was no established neoadjuvant chemotherapy available while, on the contrary, patients with stable Gastrointestinal stromal tumours (GIST) were maintained on Imatinib treatment with the postponement of their operations, allowing higher priority cases to be done [13].

The American Gastroenterological Association (AGA) also released guidance on how to implement a triage system, as illustrated in Figure 3. They advised that the “proposed framework of separating procedures into time-sensitive and non–time-sensitive cases may be useful in determining which procedures, if delayed, may negatively impact on patient-important outcomes”. They deliberately focussed on patient-important outcomes to help make decisions on triaging as it can be difficult in the current climate to categorise procedures as “elective vs non-elective” [14].

## 4. Oncological Management—A Paradigm Shift

Oncology departments have had to make fundamental changes to the way cancer care is delivered. The National Institute for Health and Care Excellence (NICE) published a COVID-19 rapid guideline on the 20th March 2020 advising on the delivery of SACT. This document offered advice on how oncologists could continue to deliver SACT, should cancer care need to be rationed or the priorities of care changed. NICE identified some key factors that should be considered, including the level of immunosuppression associated with the treatment, patient-related factors, such as the type and stage of cancer and chance of success of the treatment, along with capacity or workforce issues (Table 1). Ultimately the aim of the guideline was to help medical professionals balance the risk of cancer being sub-optimally treated against the risk of the patient being dangerously immunosuppressed and succumbing to COVID-19. NICE also recommended that these prioritisation discussions should be individualised for each patient, should involve the multidisciplinary team and both the decision and the clinical reasoning leading to the decision be clearly recorded. Thereafter, the decision is to be communicated to the patient.

In addition to the prioritisations advised above, the COVID-19 pandemic has forced us to rapidly develop novel patient pathways involving newly designed risk assessments, protocols and guidelines to ensure safe delivery of treatments including chemotherapy, immunotherapy and targeted treatments. Clinics changed almost overnight, from face-to-face reviews to contacting patients at home via telephone or videoconference calls, as huge efforts were made to decrease footfall into hospitals and outpatient departments with the aim of protecting patients and staff from contracting COVID-19 via nosocomial spread. 

In our department, all patients are contacted and assessed using a pre-screening questionnaire as to their COVID risk, they also have a COVID swab test prior to every cycle of chemotherapy. Likewise, all oncology staff in our department undergo weekly COVID testing. The oncology ward and chemotherapy day unit are deemed “clean” areas—no known COVID positive patients are allowed to enter, staff must wear personal protective equipment in line with infection control policies. The hospital suspended all visitations from relatives, apart from patients who are dying. All entrances to the hospital are secured and fronted with staff who perform temperature checks on every person entering, as well as providing everyone with a surgical mask to be worn. Patients found to have a temperature at screening or risks of infection on the screening questionnaire are not be allowed to enter the hospital; unwell patients get transferred to designated isolation areas. These measures, similar to those that were advised by the National Cancer Center/Cancer Hospital, Chinese Academy of Sciences at the start of the pandemic for Chinese hospitals [15], were introduced during the first phase of the pandemic and remain in place today.

Patients and clinical staff have adapted well. Patients, particularly those who would have had to travel far to attend appointments are saving on travel time, cost and the stress of finding hospital parking and have welcomed having clinic reviews from the comfort of their own homes. Video calls retain the ability of the oncologist to “see” the patient and are superior to telephone consultation. 

However, IT constraints, be it the patient’s or the hospital’s systems, can be a limiting factor. In addition, some patients may not be so forthcoming on the telephone and the lack of face-to-face contact means clinical examinations cannot occur. Conveying bad news or conversations around making difficult decisions are less suited to video or telephone consultation; but nowadays challenging face-to-face discussions with a patient attending alone (due to severe restrictions on visitors attending hospital), are also not ideal for the individual.

Overall, there has been a genuine paradigm shift within oncology; it remains to be seen whether these changes will remain once the pandemic is over. 

## 5. Discussion

### 5.1. COVID-19 and Cancer Correlation

Despite initial concerns about negative association between COVID-19 and cancer, emerging data have failed to demonstrate any strong correlation [16,17]. Data from the USA highlighted that cancer patients were not at an increased risk from adverse effects of COVID-19, and administration of cytotoxic chemotherapy was not significantly associated with a severe or critical COVID-19 event (*n* = 309 cancer patients, HR, 1.10; 95% CI, 0.73 to 1.60) [18]. Likewise, studies from the UK also reported that chemotherapy in the past 4 weeks had no effect on mortality from COVID-19 (OR 1·18 (0·81–1·72)) (*n* = 800 cancer patients) [19]. 

Our department published data on the outcomes of cancer patients infected with COVID-19, compared against non-cancer patients infected during the same time period. This was a collaboration of five North London hospitals comparing the outcomes of cancer patients infected with COVID-19, (Cohort A), against patients admitted with COVID-19, who did not have cancer, (Cohort B) [20]. The cohorts were matched in terms of age, sex and co-morbidities. The median age in both cohorts was the same—74 years, with 67% of patients being male. Thirty patients had a cancer diagnosis, and ninety patients did not, resulting in a 1:3 ratio. They found that the odds ratio (OR) for mortality, comparing patients with cancer to those without, was 1.05 (95% confidence interval (CI) 0.4–2.5). The severe outcome reported at (OR 0.89, 95% CI 0.4–2.0). This suggested no increased risk of death or a severe outcome in patients with cancer. Having a cancer diagnosis did not appear to increase the risk of death or severe outcome for those patients who contracted COVID-19 when compared with those infected with the virus but who were cancer free.

A subsequent study, also from our department entitled, CAPITOL (COVID-19 Cancer PatienT Outcomes in North London), analysed 2871 patients receiving SACT from 2 March to 31 May, during which time 68 patients (2.4%) contracted COVID-19. Patients undergoing SACT who then contracted the virus were more likely to die than those who did not with (adjusted (adj.) OR 9.84; 95% CI 5.73–16.9). Receiving chemotherapy increased the risk of developing COVID-19 (adj. OR 2.99; 95% CI = 1.72–5.21), with high dose chemotherapy significantly increasing risk (adj. OR 2.36, 95% CI 1.35–6.48). The presence of comorbidities also increased the risk (adjusted OR 2.29; 95% CI 1.19–4.38) as well as having a respiratory or intrathoracic neoplasm (adj. OR 2.12; 95% CI 1.04–4.36). Patients receiving targeted treatment had a surprisingly protective effect (adj. OR 0.53; 95% CI 0.30–0.95). There was no significant effect on the risk when treatment intent (curative versus palliative), hormonal therapy or immunotherapy and solid versus haematological cancers was assessed. This retrospective study cited small sample size and heterogeneity of tumour types as limitations of the study but concluded that old age and multiple co-morbidities were risk factors that adversely affected cancer patients infected with COVID-19, providing clinically meaningful information at a time when little was known [21]. 

Large Imperial-led large pan-European collaboration (*n* = 890 patients) provided a snapshot of COVID-19 pandemic in European cancer patients. Greater than 50% of cancer patients in the study developed or presented with complications from COVID-19 and in some cases associated mortality rates were greater than 70%. Due to these findings, the authors attempted to identify clinical predictors of severe COVID-19 disease. These subsets of patients demonstrating particularly adverse outcomes shared certain characteristics, including male gender, age ≥65 and the presence of ≥2 comorbidities prior to SARS-CoV-2 infection. Such patients developed severe infection with complications and, interestingly, this outcome was not related to tumour stage or the presence of measurable disease. Intriguingly, the study showed that active anticancer treatment was not associated with worse mortality. Of importance, however, is that patients on treatment who had lower mortality also had more favourable characteristics, such as being younger, having less comorbidities and being female [22]. 

The SOAP-study (Sars-CoV-2 fOr cAncer Patients), which ran alongside the COVID-ImmunoPhenotyping (COVID-IP), demonstrated that the COVID-19 signature for solid cancer patients had strikingly similar appearances to the COVID-19 signature of subjects infected with COVID-19 without cancer. Some properties were exaggerated in COVID-19 positive solid cancer patients, but the majority of patients cleared the virus, recovered from COVID-19, and re-established immunological status quo, including several severely ill stage IV cancer patients. In addition, their observations, along with other studies, suggests that the main cause of elevated mortality risk from COVID-19 and increased severity of COVID-19 in solid cancer individuals, is cancer progression. They concluded that poor COVID-19-associated cancer outcomes probably reflect the inferior cancer care resulting from public health measures to protect individuals from COVID-19. This is supported by the fact that recovered solid cancer patients showed no major immunological legacy. In comparison, COVID-19 positive haematology cancer patients showed a striking expressions of exhaustion markers by CD8T cells, which are thought to compromise virus clearance. This is consistent with the observation that “>70% of COVID-19+ haem cancer patients studied displayed prolonged viral persistence” [23]. Other studies have highlighted that patients with haematological malignancies have a poorer outcome than those with solid tumours [16,24,25].

It is imperative we use the available evidence base when deciding treatment strategies, which should be discussed openly with our cancer patients; offering anti-cancer treatment is safe, but the individualized risk to the patient should be assessed [22,26,27]. 

### 5.2. Can We Predict Patients Most at Risk from COVID-19 Complications?

A collaborative group from hospitals and academic centres in Northern England, including The Christie NHS foundation trust and The University of Manchester, have developed a risk evaluation tool, CORONET (COVID-19 Risk in Oncology tool), which is freely available online. The tool aims to identify cancer patients who are at low risk of COVID-19 complications compared to those at high risk of severe complications, to aid clinical management decisions. Low risk patients could avoid hospital admission and potentially be managed at home or via an ambulatory care system, whereas high risk patients would definitely need hospitalisation and are flagged as more likely to suffer serious infection with the need for oxygen and/or intensive care. The risk evaluation tool has been derived from interrogating real-world data covering clinical, haematological and biochemical features from UK cancer patients infected with COVID-19. Multivariable logistic regression and random forest models were used to generate the model, which was then validated and used to design the risk evaluation tool. The work is ongoing and awaiting peer review prior to publication so is not yet recommended for patient use; however, it has the potential to be a useful clinical prediction tool [28]. 

### 5.3. Cancer and COVID-19 Research Summit: Can Research Be Leading the Way Out?

The National Cancer Research Institute (NCRI), Cancer Research UK (CRUK) and Public Health England (PHE) hosted a cancer and COVID-19 research summit on 19 November. This involved researchers from varied scientific backgrounds discussing the impact COVID-19 has had, and is having, on cancer services and, equally as importantly, on cancer research. Researchers working on cancer and COVID-19-related projects were brought together to try and direct research efforts. The impact of COVID-19 on cancer prevention, early diagnosis and screening, as well as impact on cancer care pathways were identified as a key research topic. There was an acknowledgment that initial policies, for example, shielding, that were enacted to protect cancer patients at the start of the pandemic, were too broad and based on assumptions of risk rather than evidence. The disruption to endoscopy services was specifically listed as a review area, suggesting that potentially diagnosable and treatable cancers would not have been detected during this time. The resultant impact of this is another important question—will there have been a stage shift in the cancer and how can endoscopy services be maintained in future waves to prevent this from happening? The summit also discussed the need for the collection and easy sharing of data to allow collaborative work between clinicians, researchers and healthcare policy makers. They suggested the formation of a trusted research environment (TRE) so health data could be shared. They are developing a Health Data Research UK (HDR UK) Innovation Gateway to signpost the healthcare data that are already available and encourage researchers to share data so that real time information is used to support clinical decision making. This summit concluded: “Researchers highlighted that policies and mitigation strategies, such as shielding advice and alterations to treatment pathways, were too broad and at times based on assumptions that lacked evidence. This likely resulted in missed cancer diagnoses and limited ability to provide cancer care. While decision-makers acted on the best knowledge available to them at the time, we now know a great deal more about Covid-19. There needs to be a clearer process for translating such findings into practice and to inform policy. This will be important in dealing with Covid-19 and other crises, as well as improve how we assess and communicate infection risk in cancer patients to mitigate the impact of any future pandemics.” [29].

## 6. Future Direction

The National Institute of Health Research (NIHR) listed “detecting cancer early” as one of the top 10 research priorities in 2019, aimed at improving survival. Unfortunately, there has been a 25% reduction in urgent cancer referrals, estimated to equate to 2300 undiagnosed cases per week [30]. Between April and August 2020, 350,000 fewer urgent referrals for suspected cancer were made in the UK. It has been estimated that there will be a 20% increase in deaths occurring over a year in England in patients with new cancer diagnoses as a result of the COVID-19 pandemic, equivalent to an additional 6270 deaths [30].

Cancer Research UK reports that there has been a reduction in seven key diagnostic tests, (endoscopy, sigmoidoscopy, flexible-sigmoidoscopy, cystoscopy, CT scans, ultrasound and MRI) by 39% between March and May 2020—this means further longer waiting times to diagnosis. Numbers of patients starting cancer treatment also fell with a 37% drop in patients starting treatment in England in May 2020 compared to May 2019 [30]. 

In the midst of the second peak and with London being both a hub of cancer research, but at the same time heavily affected by COVID-19 pandemic, the city may face more significant challenges to come. We must ensure cancer pathways are both flexible, responsive and adaptable, yet robust so that both diagnostics services and surgery continue without interruption; especially now we have data suggesting cancer patients on treatment are not at increased risk of adverse effects from COVID-19 (Figure 4). 

In addition, another platform to consider includes public health messaging which can change the public’s perception of the risk of contracting COVID-19 versus the risks of the avoidance of seeking medical advice if they are experiencing symptoms suggestive of cancer. There are already effective and moving cancer awareness campaigns portrayed in the media, targeted towards patients who may be apprehensive about seeking advice but also concerned about symptoms they may have, i.e., the “NHS Just campaign”. In the current time, where the vaccination program is underway and we are seeing a drop in the number of new cases of COVID-19, there should also be sufficient amounts of evidence-based information for health-care workers so that they can adequately balance the risks of appropriately assessing patients with suspected cancer, assess the risks and benefits of starting immunosuppressive treatment, assessing the risks and benefits of undertaking diagnostic procedures and there is also a strong need to consider options for increasing diagnostic and treatment capacity. 

At the time of writing this review, we established that the UK was one of the first countries to have reported on a variant of SARS-CoV-2, a virus with 70% more potential to spread. The hospitals in London are particularly deemed to be most susceptible to unprecedented pressures. The senior medical adviser at Public Health England (PHE), acknowledged the rapid rise and extreme concern associated with new surge in the virus and stated: “We are continuing to see unprecedented levels of Covid-19 infection across the UK, which is of extreme concern, particularly as our hospitals are at their most vulnerable”. Unfortunately, this will inevitably have an effect on already-compromised cancer care; however, science is likely to find its way out of this pandemic, as it did in 1918.

## 7. Conclusions

The development and release of the vaccine signifies a remarkable achievement in science and delivers hope that we will soon be able to return to more normal times. However, news of a new mutant strain that is more highly contagious is extremely worrying and suggests the epidemic is far from over. However, the delivery of the vaccines has proved promising and we will need to adapt in order to deliver cancer care alongside the ongoing epidemic. With the seismic changes that have occurred within cancer care as a result of the pandemic, we should seize this opportunity as a foundation upon which a more efficient and effective care delivery model can emerge. The Oncology community now needs to adapt to the presence of two global burdens—cancer and COVID. The evidence thus far, detailing the outcomes of cancer patients in the COVID-19 pandemic, is relatively reassuring and suggests that safe cancer care provision without dramatically affecting the outcomes is indeed possible.

## Figures and Tables

**Figure 1 cancers-13-00397-f001:**
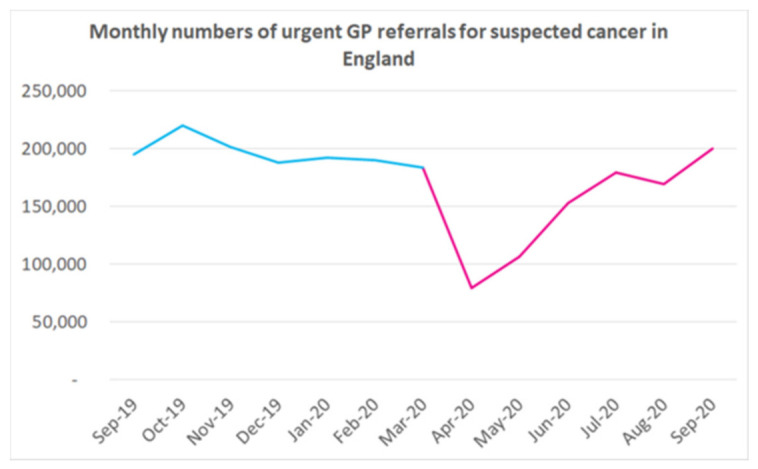
Courtesy of CRUK—https://www.cancerresearchuk.org/health-professional/diagnosis/hp-covid-19-and-cancer-hub#HP_COVID-190.

**Figure 2 cancers-13-00397-f002:**
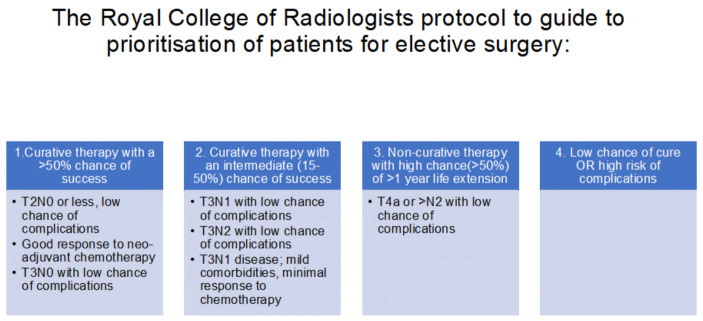
The UGI surgeons at The Royal Marsden hospital, London, collaborated with the Association of Upper Gastrointestinal Surgeons (AUGIS) and NHS England Specialty Guide for Cancer and Coronavirus and devised a way of clinically prioritising patients for UGI surgery. This framework allowed surgery to continue and was further refined as they became the West London specialist surgery “cancer hub”. They focused on effective triage, via the discussion of each case at specialist UGI multidisciplinary team meetings (sMDT) to personalise treatment plans according to clinical state, disease stage and whether alternative treatments were possible. Surgical cases were triaged according to clinical need.

**Figure 3 cancers-13-00397-f003:**
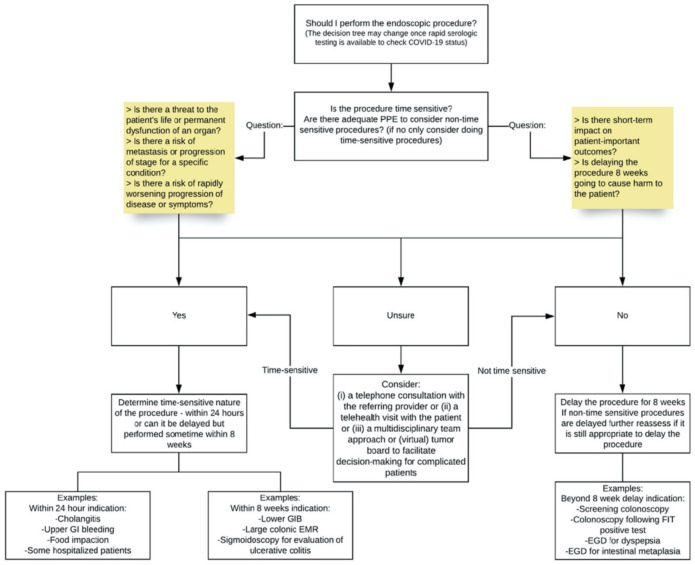
Triage system for endoscopy in the COVID-19 pandemic.

**Figure 4 cancers-13-00397-f004:**
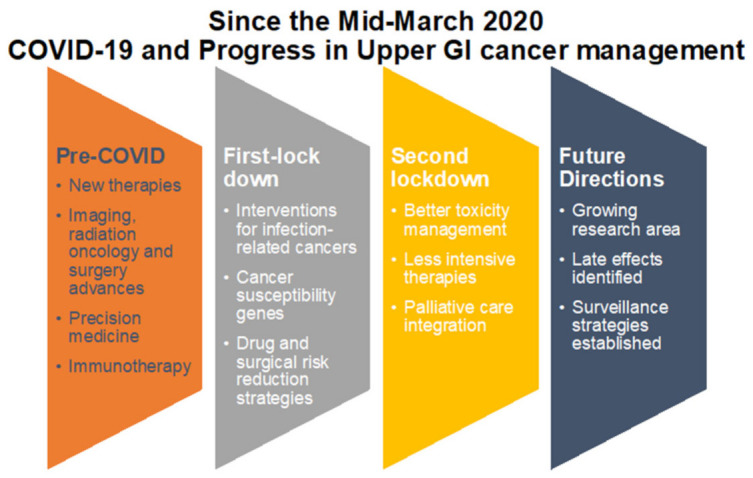
A key priority appears to be succinct triage of patient cases. Extra resources, for example, logistical, administrative or IT services, may be needed to support this clinical activity to both ensure high risk cases are managed appropriately and to avoid using finite resources on low risk cases that could wait. Close, collaborative working between all members of already existing MDTs will be pivotal in addressing the backlog and to manage the expected increase in demand once cancer referral rates return to normal.

**Table 1 cancers-13-00397-t001:** Prioritising systemic anticancer treatments. Taken from NICE guideline [NG161].

Priority Level	Treatment
1	Curative treatment with a high (more than 50%) chance of success. Adjuvant or neoadjuvant treatment which adds at least 50% chance of cure to surgery or radiotherapy alone or treatment given at relapse.
2	Curative treatment with an intermediate (20% to 50%) chance of success. Adjuvant or neoadjuvant treatment which adds 20% to 50% chance of cure to surgery or radiotherapy alone or treatment given at relapse.
3	Curative treatment with a low (10% to 20%) chance of success. Adjuvant or neoadjuvant treatment which adds 10% to 20% chance of cure to surgery or radiotherapy alone or treatment given at relapse. Non-curative treatment with a high (more than 50%) chance of more than 1 year extension to life.
4	Curative treatment with a very low (0% to 10%) chance of success. Adjuvant or neoadjuvant treatment which adds less than 10% chance of cure to surgery or radiotherapy alone or treatment given at relapse. Non-curative treatment with an intermediate (15% to 50%) chance of more than 1 year extension to life.
5	Non-curative treatment with a high (more than 50%) chance of palliation or temporary tumour control and less than 1 year expected extension to life.
6	Non-curative treatment with an intermediate (15% to 50%) chance of palliation or temporary tumour control and less than 1 year expected extension to life.

https://www.nice.org.uk/guidance/ng161/chapter/6-Prioritising-systemic-anticancer-treatments.

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
