# Peer review of "COVID-19 and Its Impact on Upper Gastrointestinal (GI) Cancer Management"

_cancers, 2021, doi:10.3390/cancers13030397_

Round 1
Reviewer 1 Report
Journal: Cancers
Manuscript ID: cancers-1081023
Title: COVID-19 and its impact on early stage upper gastrointestinal (GI) cancer management
Authors: Fernando S, et al.
This quick review article try to summarize and consider the impact of COVID-19 pandemic on oncology and its future perspectives, especially of cancer management for upper GI cancer patients.
In addition to “controversies on COVID-19, and management and outcomes of cancer patients in 5th section, I would like to read your present best practice or recommendations for diagnosis and treatment of GI cancer referring to reports from international collaborative consensus, AGA, and so on. I also prefer to have a short comment on post-COVID syndrome in cancer patients if possible, because this is one of upcoming issues.
Some minor corrections:
From the content of the paper, wording in the title of “early stage upper gastrointestinal (GI) cancer” appears to be inappropriate and may be just “upper gastrointestinal (GI) cancer”
In Fig. 1, the period of first as well as second lockdown should be indicated in the figure.
“Considerations for treatment of oesophagogastric cancers within the United Kingdom during the Covid-19 pandemic” in Page 3 and “a risk evaluation tool, CORONET (COVID-19 Risk in Oncology tool), which is freely available online” in Page 6 had better to be supplied with their web sites or references.
Abbreviation in initial appearance may require full-spelling, e.g., MTD and ITU.
“Gastrointestinal stromal cell tumours (GIST)” may be “Gastrointestinal stromal tumours (GIST)”
Author Response
Response to Reviewers/Editor
We thank the three reviewers for their very useful comments on our manuscript entitled COVID-19 and its impact on early stage upper gastrointestinal (GI) cancer management. We have addressed each of the reviewer’s comments below.
Reviewer 1:
In addition to “controversies on COVID-19, and management and outcomes of cancer patients in 5th section, I would like to read your present best practice or recommendations for diagnosis and treatment of GI cancer referring to reports from international collaborative consensus, AGA, and so on. I also prefer to have a short comment on post-COVID syndrome in cancer patients if possible, because this is one of here:upcoming issues.
These have been addressed: The AGA recommendations have been addressed in line 178 with a figure at line 185. Effects post-COVID on Oncology patients have also been addressed starting at line 297.
Some minor corrections:
From the content of the paper, wording in the title of “early stage upper gastrointestinal (GI) cancer” appears to be inappropriate and may be just “upper gastrointestinal (GI) cancer”
We have changed the title after reflecting on this comment from reviewer 1 (line 47)
In Fig. 1, the period of first as well as second lockdown should be indicated in the figure.
Thank you very much for this comment- unfortunately these data are only emerging and CRUK have not provided access to data on impact of lock down following the 2nd wave of COVID-19.
“Considerations for treatment of oesophagogastric cancers within the United Kingdom during the Covid-19 pandemic” in Page 3 and “a risk evaluation tool, CORONET (COVID-19 Risk in Oncology tool), which is freely available online” in Page 6 had better to be supplied with their web sites or references.
The weblink to the RCR document mentioned above in page 3 has been added to the bottom of the page.
CORONET risk evaluation tool is referenced, reference 28. An online link to the website has also been added at the end of the references.
Abbreviation in initial appearance may require full-spelling, e.g., MTD and ITU.
We have now corrected this, using the full spelling at first appearance (line 51, line 280)
“Gastrointestinal stromal cell tumours (GIST)” may be “Gastrointestinal stromal tumours (GIST)”
We have now corrected this (line 130-131)
Reviewer 2 Report
The paper is well written and reports recent information. Minor objections are:
- Some abbreviations have not been fully explained (e.g. ITU, page 1)
- Figure 1: do the reported value represent the raw number or the rate standardized for total population?
- Despite the title mentions "early stage upper gastrointestinal (GI) cancer", which are cancers that may be treated by endoscopy without oncologic therapy (i.e. early gastric cancer, dysplastic Barrett/T1a esophageal cancer...), this point was not fully discussed. Please amend.
Author Response
Response to Reviewers/Editor
We thank the three reviewers for their very useful comments on our manuscript entitled COVID-19 and its impact on early stage upper gastrointestinal (GI) cancer management. We have addressed each of the reviewer’s comments below.
Reviewer 2:
Some abbreviations have not been fully explained (e.g. ITU, page 1)
We have now corrected this, as above, after same comment from reviewer 1 (line 51).
Figure 1: do the reported value represent the raw number or the rate standardized for total population?
The value represents the raw value.
Despite the title mentions "early stage upper gastrointestinal (GI) cancer", which are cancers that may be treated by endoscopy without oncologic therapy (i.e. early gastric cancer, dysplastic Barrett/T1a esophageal cancer...), this point was not fully discussed. Please amend.
We have now corrected this, as above, after same comment from reviewer 1 (line 47).
Reviewer 3 Report
In my opinion, the overall level of the paper is good structured: it is well written and some important considerations are highlighted. The conclusions sections provide useful information for the readers mainly emphasizing the suggestion for the oncologists to adapt to the presence of two global burdens - cancer and COVID.
Some minor changes should be performed:
Please check the few spelling grammatical error in the text.
Author Response
Reviewer 4
The article was revised on 14th January 2021 (when the latest version was sent through) and any spelling/grammatical errors have been corrected, as far as we can see.
Reviewer 4 Report
Fernando et al present a timely and up-to-date overview of the impact of COVID-19 on the diagnosis and management of cancer, in particular upper gastrointestinal cancer in the UK.
Find below some minor comments for review. Also, check the manuscript for minor spelling errors.
Subheading 1:
- Define ITU when first used in the introduction
- Do you mean “underpowered” instead of “unpowered”? Pleas revise
Subheading 2:
- Is the following statement from your own data or does it need a reference? Can you provide numbers for this statement? It is an interesting observation, which would have more meaning if you can provide numbers.
“Moreover, oesophageal and gastric cancers presented late and were more likely to be of advanced stage at diagnosis potentially leading to adverse outcomes.”
Subheading 3:
- Change to past tense: “Liver resections for management of hepatocellular carcinoma (HCC) went ahead given there is no established neoadjuvant chemotherapy available
Subheading 4:
- Please revise the following sentence and remove “that”:
NICE also recommended these prioritisation discussions should be individualised for each patient, should involve the multidisciplinary team and that both the decision and the clinical reasoning leading to the decision be clearly recorded
- In the following paragraph, the author switches between present and past tense. Are these measures still in place or are we talking about measures in place during the first wave. Please revise accordingly:
“In our department, all patients are contacted and are assessed using pre-screening questionnaire as to their COVID risk, they also have a COVID swab test prior to every cycle of chemotherapy. Likewise, all oncology staff in our department undergo weekly COVID testing. The oncology ward and chemotherapy day unit are deemed ‘clean’ areas - no known COVID positive patients are allowed to enter, staff must wear personal protective equipment in line with infection control policies. The hospital suspended all visitations from relatives, apart from patients who are dying. All entrances to the hospital were secured and fronted with staff who perform temperature checks on every person entering, as well as providing everyone with a surgical mask to be worn. Patients found to have temperature at screening or risks of infection on screening questionnaire would not be allowed to enter the hospital; unwell patients being moved to designated isolation areas.”
Subheading 5:
- Do you mean “five” in the following phrase:
“between fiver North London hospitals”
- Can the author change the following to “matched” instead of “matches”:
“In this age, sex and comorbidities matches study”
Subheading 6:
- Potentially revise and update to current situation:
“In the current time, where we are now over the second lockdown and vaccinations are looming over the horizon, there should also be sufficient amounts of evidence-based information”
- Potentially revise and update to current situation:
“At the time of writing this review, we established that UK was one of the first countries to have reported on a variant of SARS-CoV-2, a virus with 70% more potential to spread.”
Author Response
Reviewer 4:
Define ITU when first used in the introduction
This has already been addressed, as above.
Do you mean “underpowered” instead of “unpowered”?
We have changed the word to underpowered (line 54)
Is the following statement from your own data or does it need a reference? Can you provide numbers for this statement? It is an interesting observation, which would have more meaning if you can provide numbers.
“Moreover, oesophageal and gastric cancers presented late and were more likely to be of advanced stage at diagnosis potentially leading to adverse outcomes.”
We thank reviewer 3 for this comment and we have now referenced this statement (reference number 11).
Change to past tense: “Liver resections for management of hepatocellular carcinoma (HCC) went ahead given there is no established neoadjuvant chemotherapy available
We have made this change (line 129)
Please revise the following sentence and remove “that”:
NICE also recommended these prioritisation discussions should be individualised for each patient, should involve the multidisciplinary team and that both the decision and the clinical reasoning leading to the decision be clearly recorded
We have removed ‘that’ (line 159)
In the following paragraph, the author switches between present and past tense. Are these measures still in place or are we talking about measures in place during the first wave. Please revise accordingly:
“In our department, all patients are contacted and are assessed using pre-screening questionnaire as to their COVID risk, they also have a COVID swab test prior to every cycle of chemotherapy. Likewise, all oncology staff in our department undergo weekly COVID testing. The oncology ward and chemotherapy day unit are deemed ‘clean’ areas - no known COVID positive patients are allowed to enter, staff must wear personal protective equipment in line with infection control policies. The hospital suspended all visitations from relatives, apart from patients who are dying. All entrances to the hospital were secured and fronted with staff who perform temperature checks on every person entering, as well as providing everyone with a surgical mask to be worn. Patients found to have temperature at screening or risks of infection on screening questionnaire would not be allowed to enter the hospital; unwell patients being moved to designated isolation areas.”
All measures mentioned started during the first wave, have been in place throughout and continue to be in place at present, so this paragraph has been made clearer.
Do you mean “five” in the following phrase: “between fiver North London hospitals”
Yes we did mean ‘five’ and this has now been changed, (line 212).
Can the author change the following to “matched” instead of “matches”:
“In this age, sex and comorbidities matches study”
We have changed to “matched”, (line 214)
Potentially revise and update to current situation:
“In the current time, where we are now over the second lockdown and vaccinations are looming over the horizon, there should also be sufficient amounts of evidence-based information”
Potentially revise and update to current situation:
“At the time of writing this review, we established that UK was one of the first countries to have reported on a variant of SARS-CoV-2, a virus with 70% more potential to spread.”
We agree with reviewer 3 and have changed the sentence to make a more up to date statement (line 356).